# Optimized Antibiotic Management of Critically Ill Patients with Severe Pneumonia Following Multiplex Polymerase Chain Reaction Testing: A Prospective Clinical Exploratory Trial

**DOI:** 10.3390/antibiotics13010067

**Published:** 2024-01-10

**Authors:** Alexia Verroken, Julien Favresse, Ahalieyah Anantharajah, Hector Rodriguez-Villalobos, Xavier Wittebole, Pierre-François Laterre

**Affiliations:** 1Department of Microbiology, Cliniques Universitaires Saint-Luc, Université Catholique de Louvain, Avenue Hippocrate 10, 1200 Brussels, Belgium; 2Department of Critical Care Medecine, Cliniques Universitaires Saint-Luc, Université Catholique de Louvain, Avenue Hippocrate 10, 1200 Brussels, Belgium; xavier.wittebole@saintluc.uclouvain.be; 3Department of Critical Care Medecine, Centres Hospitaliers Universitaires HELORA, 1200 Brussels, Belgium

**Keywords:** severe pneumonia, molecular testing, FilmArray Pneumonia, antibiotic management, critically ill, antimicrobial stewardship

## Abstract

Molecular diagnostic testing is assumed to enable fast respiratory pathogen identification and contribute to improved pneumonia management. We set up a prospective clinical trial at a tertiary hospital intensive care unit including adult patients suspected of severe pneumonia from whom a lower respiratory tract sample could be obtained. During control periods (CPs), routine testing was performed, and during intervention periods (IPs), this testing was completed with the FilmArray Pneumonia Panel *plus* test (FA-PNEU) executed 24/7. The main objective was to measure the impact of FA-PNEU results in terms of reduced time to targeted antimicrobial treatment administration. Over a 10-month period, analysis was performed on 35 CP and 50 IP patients. The median time to targeted antimicrobial treatment administration was reduced to 4.3 h in IPs compared to 26.4 h in CPs, with 54% of IP patients having FA-PNEU results that led to a treatment modification, of which all but one were targeted. Modifications included 10 (37%) de-escalations, 7 (25.9%) escalations, 3 (11.1%) regimen switches, and 7 (25.9%) complete antimicrobial discontinuations. FA-PNEU results were available with a 42.3 h gain compared to routine identification. This prospective study confirmed retrospective data demonstrating the benefit of FA-PNEU testing in severe pneumonia management of critically ill patients through improved antimicrobial use.

## 1. Introduction

Critical care physicians manage patients with various types of lung infection such as community-acquired pneumonia (CAP), hospital-acquired pneumonia (HAP), and ventilator-acquired pneumonia (VAP), all of them being associated with high morbidity and mortality [1,2]. International guidelines strongly recommend the initiation of empiric therapy for patients suspected of severe pneumonia and provide an algorithm towards the orientation of narrow- or broad-spectrum antibiotics according to the risk of multidrug-resistant pathogens and mortality [3,4]. Subsequent tailoring would then be possible according to respiratory sample culture results around day 2–3. Rapid molecular testing has the ability to drastically reduce this timeslot, enabling targeted antimicrobial treatment (TAT) to be administered more rapidly and limiting the use of broad-spectrum antibiotics and its consequences of increased toxicity, high costs, and antimicrobial resistance selection [5]. For several years, multiplex PCR approaches have been made available and offer sped-up results compared to microbiological routine testing with even better microorganism recovery and co-infection detection [6,7,8]. Observational studies further suggest improved pneumonia management of intensive care unit (ICU) patients through faster implementation of the optimal treatment, yet these conclusions have to be validated by prospective interventional studies [9,10,11]. We therefore set up a clinical exploratory trial with the aim of quantifying the impact of molecular diagnostic testing results on antibiotic administration in ICU patients with suspected pneumonia.

## 2. Results

Over a 10-month period, 104 ICU patients with a suspicion of severe acute pneumonia were enrolled, yet a lower respiratory sample was lacking for 11 of them. Subsequently, 93 patients were endorsed during the CPs (*n* = 37) and IPs (*n* = 56 patients). Ultimately, analysis was performed on 35 patients in the CPs and 50 patients in the IPs after exclusion due to early death and erroneous allocation.

CP and IP populations were statistically comparable in terms of age, sex ratio, severity scores, comorbidities (except for cardiovascular disorders), and final diagnosis, as detailed in Table 1. CAP was the main final diagnosis, representing 51.4% and 48% of the included CP and IP patients, respectively. HAP and VAP combined concerned 28.6% of CP patients and 26% of IP patients. The main pneumonia sources were bacterial (78.6% in CPs and 62.2% in IPs) and viral (7.1% in CPs and 13.5% in IPs). Pneumonia diagnosis was reasonably excluded following negative microbiological results in 20% of CP patients and 26% of IP patients.

Figure 1 summarizes the median time to microbiological results and median time to the administration of TAT. FA-PNEU results were made available within a median time of 2.5 h (*p* < 0.001) following sample registration in the laboratory, while routine ID results required an extended median time to results of 45.2 and 44.8 h, respectively, in CPs and IPs (*p* = 0.5618). The median time to TAT administration was reduced to 4.3 h in IPs compared to 26.4 h in CPs (*p* = 0.0376).

The direct communication of FA-PNEU results in IPs led to sped-up antimicrobial modifications in 27/50 (54%) patients, as detailed in Table 2. Modifications included 10 (37%) de-escalations, 7 (25.9%) escalations, 3 (11.1%) regimen switches, and 7 (25.9%) complete antimicrobial discontinuations. All treatments initiated following FA-PNEU results were TAT, except for one patient for whom vancomycin was initiated following the FA-PNEU detection of a ≥10^7^ copies/mL *Staphylococcus aureus* combined with the *mecA*/*C* and MREC resistance gene. For the latter, routine testing led to the monomicrobial growth of numerous oxacillin-susceptible *S. aureus* colonies. The patient was therefore ultimately treated with flucloxacillin.

28-day mortality in IPs concerned 10/50 (20%) patients and 14/35 (40%) patients in CPs. The median ICU length of stay was 3 days and 21 h and 5 days and 3 h in, respectively, IPs and CPs. *p*-values were not statistically significant for the latter outcomes.

A comparison of FA-PNEU and IP patient routine testing results for the qualitative analysis of the 15 on-panel typical bacteria led to an overall PPA of 100% (34/34) and NPA of 97% (685/706). The main bacteria exclusively detected by FA-PNEU were *H. influenzae* (9), *S. pneumoniae* (3) and *S. aureus* (3). During IPs, routine analysis retrieved six respiratory viruses. FA-PNEU testing also detected the latter respiratory viruses, yet it identified four additional viruses not requested through routine testing. Four of the five resistance genes detected with FA-PNEU were in concordance with the phenotypical AST results. However, one detection of the *mecA*/*mecC* + MREJ marker was discordant with a culture-growing oxacillin-susceptible *S. aureus*.

## 3. Discussion

In this clinical exploratory trial, 52% of all IP ICU patients benefited from an accelerated TAT initiation following FA-PNEU results with a global median time reduction of 22.1 h towards TAT compared to CP patients. Similarly, in a retrospective study on adult hospitalized patients with lower respiratory tract infections, Buchan et al. measured potential for appropriate antimicrobial modifications following FA-PNEU results in 52.5% of the evaluable patients, yet they also reported inappropriate modifications in 18.2% [8]. In another retrospective multi-center study from Monard et al., early adaptation of antimicrobial therapy following FA-PNEU testing was measured at 77%, mainly consisting of de-escalation [10]. In our study, de-escalation and total treatment discontinuation accounted for 62.9% of all modifications following FA-PNEU results. The latter observations hereby strongly emphasize the usefulness of rapid molecular testing in the reduced use of broad-spectrum antibiotics. A Greek ICU team confronted on a daily basis with high rates of multidrug-resistant pathogens detected significant broad-spectrum antibiotic savings following the implementation of rapid molecular testing of severe CAP, HAP, or VAP among ICU patients [11]. Sped-up antibiotic tailoring is of major importance as it has been widely demonstrated that the use of broad-spectrum antibiotics also entails adverse effects. An impressive 20% increase in the odds of death was calculated among septic patients receiving unnecessary broad-spectrum antibiotics in an extensive cohort study including 17,430 culture-proven septic patients operated by Rhee et al. [12]. With caution for possible confounding factors, the authors partially explained this higher mortality rate by increased adverse effects, acute kidney injury, and *Clostridium difficile* infections. Conversely, inadequate empiric treatment was also associated with 20% higher odds of death, confirming the urgent need for rapid diagnostic tests. A Danish team performed a prospective randomized evaluation of FA-PNEU testing in guiding the treatment of patients suspected of CAP at an emergency unit [13]. Interestingly, despite an increase in more targeted antibiotic prescription, FA-PNEU testing did not impact the prescription rate of no or narrow-spectrum antibiotics 4 h after patient admission. Data analysis further failed to identify any difference in relation to ICU admissions, re-admissions within 30 days, length of stay, 30-day mortality, and in-hospital mortality. The discussed explanations were the very low national antimicrobial resistance level and the low number of events considering mortality and ICU transfer hampering the measurement of significant outcomes. Similarly, in our study, 28-day mortality and ICU length of stay were not statistically different between IPs and CPs. Yet, the latter outcomes need to be considered with caution as the sample size is limited.

The practical set-up of rapid molecular diagnostic testing of severe pneumonia is a key component to consider during implementation. In our setting, FA-PNEU testing was performed 24/7, allowing a median time to results of 2.5 h starting from laboratory sample registration. Similarly, Crémet et al., performing FA-PNEU testing upon the sample’s laboratory arrival, reported a median turnaround time from sample collection to results of 4.3 h [6]. However, not all laboratories have the required manpower at all times nor are they necessarily in the same geographical location as the hospital. Point-of-care molecular testing has extensively proven its efficacy in emergency settings with significant reductions in antibiotic use, lengths of stay, and timely antiviral use through the rapid detection of influenza and SARS-CoV-2 [14,15]. Moreover, the latter observations could trigger a reflection on safely broadening molecular testing at the ICU patient’s bedside with severe pneumonia syndromic testing, knowing the short and straightforward sample preparation of the available commercial approaches.

FA-PNEU performances in our study were very similar to ones previously observed in various evaluation studies equally reporting >95% PPA and NPA and underlining culture-negative FA-PNEU detection of a consistent number of *S. aureus* and *Haemophilus influenzae* [8,16]. Common explanations of these discordances were low strain concentration, empirical antibiotic coverage of detected bacteria preceding sample collection, and the presence of “normal oral flora” possibly obscuring strain culture detection, hereby stressing the added value of FA-PNEU testing in the latter circumstances. Attention must be paid to the detection of the *mecA/C* and MREC resistance genes since false-positive detections have also been reported by other teams [6,16,17].

A regularly reported drawback of molecular testing is the qualitative detection of DNA rather than the numbering of viable microorganisms. To partially overcome this issue, FA-PNEU bacterial detection comes with semi-quantitative results, enabling the clinician to distinguish between colonizing strains and clinically relevant pathogens. Several teams suggested a positivity threshold below which the clinician was advised not to consider the identified bacteria for antimicrobial treatment [6,11]. Defining these thresholds should be part of a set of guidelines helping clinicians in the interpretation of molecular diagnostic results, knowing that antimicrobial stewardship programs are absolutely essential to reach significant clinical outcomes and cost-effectiveness for molecular tools [18,19,20].

Alongside the clear benefits of rapid molecular pneumonia diagnostic tests, consideration needs to be given to the inconveniences. First, these molecular approaches allow the detection of a large yet limited panel of respiratory pathogens. As an example, the evaluated FA-PNEU fails to detect potential nosocomial pneumonia bacterial pathogens including *Citrobacter koseri*, *Morganella morganii*, and *Stenotrophomonas maltophilia*. A very recent review article by Moy et al. performed a meta-analysis on the diagnostic performances of FA-PNEU for the detection of respiratory bacterial pathogens in 8968 respiratory specimens. They concluded that 9.3% of bacteria detected in a culture were not included in the panel [21]. Similarly, these tests detect broad-spectrum resistance genes that clearly do not cover all currently encountered beta-lactamases. As a result, lower respiratory tract sample cultures, the identification of significantly growing bacterial colonies, and phenotypical antimicrobial susceptibility testing remain essential. Secondly, the high cost of rapid molecular systems and their cartridges is a major barrier to their clinical routine diagnostic implementation. It is assumed that certain outcomes, such as reduced antibiotic consumption or a reduced length of stay, could compensate for the molecular testing investment; however, the valuable literature remains very limited and heterogeneous in its conclusions, hereby highlighting the urgent need for broad and well-designed randomized controlled trials [22,23].

Our study’s main drawback is the limited number of included patients. Initiated in 2019, the study was interrupted after a 10-month period following the meteoric onset of the COVID-19 pandemic. The limited number of patients in this study also resulted in its statistical weakness to significantly measure additional patient outcomes such as ICU lengths of stay and mortality rates. Similar yet broad randomized controlled trials are ongoing and intend to give a clear picture on the latter outcomes [24,25].

In summary, this preliminary prospective randomized trial confirmed retrospective data by demonstrating the benefit of FA-PNEU testing in severe pneumonia management of ICU patients and highlight the advantages and inconveniences to consider prior to implementation. Additional clinical impact and cost-effectiveness studies should enable the widespread integration of rapid molecular testing in the routine management of severe pneumonia.

## 4. Materials and Methods

### 4.1. Study Design and Setting

The prospective trial was conducted at the Cliniques universitaires Saint-Luc, a tertiary Belgian hospital, with a 22-bed medical surgery ICU. Between June 2019 and March 2020, all critically ill adult patients suspected of severe acute pneumonia were considered for enrolment, yet exclusively those from whom a lower respiratory tract sample could be obtained were ultimately included.

The control (CPs) and intervention periods (IPs) followed each other for consecutive 2-week periods. During the CPs, routine semi-quantitative bacterial cultures were performed followed by identification (ID) and antimicrobial susceptibility testing (AST) on all relevant strains.

### 4.2. Microbiological Testing of the Lower Respiratory Tract Samples

Routine diagnostic testing for viruses (immunofluorescence) and atypical bacteria (molecular) as well as urinary antigen testing was left to the clinician’s discretion.

During the IPs, additional molecular testing was performed 24/7, and results were immediately provided electronically and by direct phone communication to the ICU clinician pursuing antimicrobial optimization. The selected molecular test was the FilmArray Pneumonia Panel *plus* test (FA-PNEU, BioFire Diagnostics, Salt Lake City, UT, USA), an automated multiplex PCR test allowing the direct detection of 15 bacteria with a semi-quantitative value, 3 atypical bacteria, 9 viruses, and 7 antimicrobial resistance genes within 1 h and 15 min. Semi-quantitative measurements were reported in 10^4^, 10^5^, 10^6^, or ≥10^7^ genomic copies/mL. A threshold at ≥10^6^ genomic copies was suggested in order to consider the detected bacteria as pathogens rather than colonizing organisms. Additionally antimicrobial stewardship guidelines were set up to guide the clinician in the selection of the most appropriate and targeted antibiotherapy following FA-PNEU results in accordance with the local resistance epidemiology. Yet, final therapeutic decisions were left to the discretion of the treating ICU clinician.

### 4.3. Measured Outcomes

The main outcome of the study was measurement of the impact of the FA-PNEU results in terms of reduced time to TAT administration in the IPs versus the CPs. The secondary outcomes were 28-day mortality and ICU length of stay. Additional data such as time to FA-PNEU results, time to ID, and time to AST results were compared between the CPs and IPs. Bacterial, viral, and resistance gene detection of the FA-PNEU was ultimately compared to available IP patient routine test results.

Clinical characteristics, comorbidities, and pneumonia data were compared for all patients included in the CPs versus IPs to evaluate the similarity of the 2 study populations. Among clinical characteristics, we registered patients’ age, sex, and severity scores including APACHE II (Acute Physiology and Chronic Health Evaluation II) and SOFA (Sequential Organ Failure Assessment) scores. Compared comorbidities included active malignancy, cardiovascular disorder, chronic lung disease, diabetes, neutropenia (<500 neutrophils/μL), and organ transplant. Among the pneumonia data, we analyzed the final diagnosis and the microbiological source.

### 4.4. Data Handling and Statistical Analysis

Statistical analyses were performed using GraphPad Prism 10.0.0 (San Diego, CA, USA). Normality of distribution was assessed using a D’Agostino–Pearson test with a log-transformation. An unpaired *t*-test and chi-squared test were used for data comparison between CPs and IPs. *p*-values < 0.05 were considered statistically significant. Concordance between FA-PNEU and routine results was evaluated through a calculation of positive and negative percentage agreement (PPA/NPA).

This study was conducted in accordance with the Declaration of Helsinki and approved by the Institutional Review Board of the Hospital-Faculty Ethics Committee Saint-Luc—UCL (National number: B403).

## Figures and Tables

**Figure 1 antibiotics-13-00067-f001:**
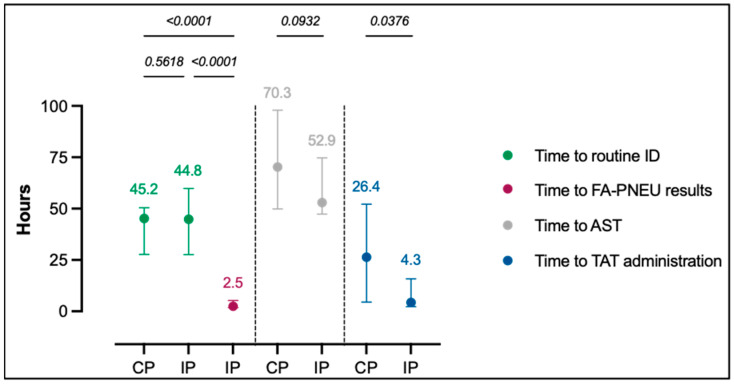
Median time to microbiological results and to TAT administration in CPs and IPs. Dots represent median time and lines on each side of the dot represent interquartile ranges. Time data written above the vertical lines are expressed in hours. *p*-values < 0.05 were considered statistically significant. AST: antimicrobial susceptibility testing, CPs: control periods, FA-PNEU: FilmArray Pneumonia Panel *plus* test, ID: identification, IPs: interventional periods, TAT: targeted antimicrobial treatment.

**Table 1 antibiotics-13-00067-t001:** Clinical characteristics and pneumonia data of patients included in the control and intervention periods.

	CPs (*n* = 35)	IPs (*n* = 50)	Total (*n* = 85)	*p*-Value
	Patient characteristics			
Age, years, mean ± SD	64.4 ± 16.2	61.4 ± 14.5	62.6 ± 15.2	0.2646
Male/female sex	23/12	31/19	54/31	0.7262
APACHE II score, mean ± SD	17.5 ± 8.9	17.9 ± 7.5	17.7 ± 8.1	0.6711
SOFA score, mean ± SD	6.5 ± 3.9	7.2 ± 3.8	6.9 ± 3.8	0.2930
	Comorbidities			
Active malignancy	7 (20)	8 (16)	15 (17.6)	0.6340
Cardiovascular disorder	12 (34.3)	28 (56)	40 (47.1)	0.0484
Chronic lung disease	8 (22.9)	10 (20)	18 (21.2)	0.7510
Diabetes	6 (17.1)	11 (22)	17 (20)	0.5817
Neutrophil count <500/μL	0	3 (6)	3 (3.5)	0.1401
Organ transplant	1 (2.9)	2 (4)	3 (3.5)	0.7787
	Final diagnosis			
Community-acquired pneumonia	18 (51.4)	24 (48)	42 (49.4)	0.7557
Hospital-acquired pneumonia	4 (11.4)	7 (14)	11 (12.9)	0.7281
Ventilation-acquired pneumonia	6 (17.1)	6 (12)	12 (14.1)	0.5028
No pneumonia	7 (20)	13 (26)	20 (23.5)	0.5210
	Microbiological source			
Bacterial	22 (78.6)	23 (62.2)	45 (52.9)	0.1254
Viral	2 (7.1)	5 (13.5)	7 (8.2)	0.4793
Mycotic	0	1 (2.7)	1 (1.2)	0.4068
Unknown	2 (7.1)	5 (13.5)	7 (8.2)	0.4793
Bacterial + viral	2 (7.1)	3 (8.1)	5 (5.9)	0.9561

Data are represented as numbers (%) unless otherwise specified. Abbreviations: APACHE II, Acute Physiology and Chronic health Evaluation II; CPs, control periods; IPs, intervention periods; SOFA, Sequential Organ Failure Assessment. *p*-values < 0.05 were considered statistically significant.

**Table 2 antibiotics-13-00067-t002:** IP ICU patients with a modified antimicrobial treatment following FA-PNEU result communication.

			FA-PNEU							
IP Patient	Age	Sexe	Final Diagnosis	Sample Type	Source	Identification	Bin (Copies/mL)	FA-PNEU Resistance Genes	Empirical Antimicrobial Treatment	Antimicrobial Treatment Following FA-PNEU Results
									Escalation	
IP21	73	Male	CAP	ETA	Bacterial	*Pseudomonas aeruginosa*	10^6^	–	ceftriaxone	ceftazidime
IP34	59	Female	CAP	ETA	Viral & bacterial	*Haemophilus influenzae*–Influenza A	≥10^7^	NA	ceftriaxone	ceftriaxone + oseltamivir
IP7	68	Male	HAP	ETA	Bacterial	*Pseudomonas aeruginosa*	≥10^7^	–	ceftriaxone	ceftazidime
IP23	79	Male	VAP	ETA	Bacterial	*Pseudomonas aeruginosa*	10^6^	NA	none	ceftazidime
IP39	58	Male	VAP	BAL	Bacterial	*Staphylococcus aureus*	≥10^7^	NA	none	flucloxacillin
IP40	61	Female	VAP	BAL	Bacterial	*Enterobacter cloacae* complex	10^4^	NDM	meropenem + amikacin	colistin + amikacin
IP46	59	Male	VAP	ETA	Bacterial	*Staphylococcus aureus*	≥10^7^	mecA/C	none	vancomycin
									De-escalation	
IP11	68	Male	CAP	ETA	Bacterial	*Klebsiella pneumoniae–Streptococcus agalactiae*	≥10^7^ & ≥10^7^	–	ceftriaxone	amoxicillin-clavulanic acid
IP17	66	Male	CAP	Sputum	Bacterial	*Streptoccocus pneumoniae*	10^6^	–	ceftriaxone	penicillin
IP24	41	Male	CAP	Sputum	Bacterial	*Streptococcus pyogenes*	≥10^7^	NA	amoxicillin-clavulanic acid	cefuroxime
IP32	57	Female	CAP	ETA	Viral & bacterial	*Haemophilus influenzae*–hMPV	10^4^	NA	ceftriaxone	cefuroxime
IP38	58	Male	CAP	ETA	Viral	Influenza A	NA	NA	piperacillin-tazobactam + oseltamivir	oseltamivir
IP4	47	Female	CAP	Sputum	Bacterial	*Haemophilus influenzae*	≥10^7^	NA	ceftriaxone	cefuroxime
IP48	86	Male	CAP	ETA	Bacterial	*Staphylococcus aureus*	≥10^7^	–	ceftriaxone	flucloxacillin
IP49	75	Male	CAP	Sputum	Bacterial	*Staphylococcus aureus*	≥10^7^	–	piperacillin-tazobactam	flucloxacillin
IP51	60	Male	CAP	ETA	Bacterial	*Streptococcus pneumoniae*	≥10^7^	NA	ceftriaxone	cefuroxime
IP14	29	Male	VAP	ETA	Bacterial	*Proteus mirabilis–Escherichia coli*	≥10^7^ & 10^6^	NA	temocillin	cefuroxime
									Regimen switch	
IP31	59	Male	CAP	Sputum	Viral	Influenza A	NA	NA	ceftriaxone	oseltamivir
IP8	76	Male	HAP	ETA	Bacterial	*Klebsiella aerogenes–Pseudomonas aeruginosa*	≥10^7^ & ≥10^7^	CTX-M	ceftriaxone	ciprofloxacin
IP10	62	Female	HAP	BAL	Mycosis *	–	NA	NA	ceftazidime	trimethoprim-sulfamethoxazole
									Antimicrobial stop	
IP2	56	Male	No	Sputum	NA	–	NA	NA	ceftriaxone	none
IP12	73	Female	No	Sputum	NA	–	NA	NA	ceftriaxone	none
IP18	79	Male	No	ETA	NA	–	NA	NA	cefuroxime	none
IP20	87	Female	No	Sputum	NA	–	NA	NA	ceftriaxone	none
IP33	89	Male	No	ETA	NA	–	NA	NA	cefuroxime	none
IP42	50	Female	No	Sputum	NA	–	NA	NA	amoxicillin-clavulanic acid	none
IP47	28	Male	No	ETA	NA	–	NA	NA	ceftriaxone	none

In this study, clarithromycin data were not reported as it was principally considered as an anti-inflammatory rather than an antimicrobial treatment. * mycosis identified through routine testing of patient IP10 was a *Pneumoncystis jirovecii*. Underlined FA-PNEU results were discordant with routine microbiological results. CAP: community-acquired pneumonia, ETA: endotracheal aspirate, FA-PNEU: FilmArray Pneumonia Panel *plus* test, HAP: hospital-acquired pneumonia, ICU: intensive care unit, IP: interventional period, NA: not applicable, VAP: ventilation-acquired pneumonia.

## Data Availability

The data generated and analyzed during this study are available upon reasonable request from the corresponding author.

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
