# Peer review of "Optimized Antibiotic Management of Critically Ill Patients with Severe Pneumonia Following Multiplex Polymerase Chain Reaction Testing: A Prospective Clinical Exploratory Trial"

_antibiotics, 2024, doi:10.3390/antibiotics13010067_

Round 1

Reviewer 1 Report

Comments and Suggestions for Authors

1.       The author claims in abstract “This pioneer study confirmed retrospective data demonstrating the benefit of FA-PNEU testing in severe……” This is not a pioneer study, already studies exist on this topic with different results.

2.       Reference 21, is unnecessarily used. The statement for delay in the study due to COVID-19 is fine but referring to your own study is not fair.

3.       In table 1, mean ± SD are used but what are the values are showing in parenthesis e.g.  male sex 23 (65.7). What is represented in the parenthesis?

4.       Table 2 is not appropriately presented. Resolution is not good. Font size can be improved.

5.       APACHE and SOFA should be described briefly in methodology.

6.       In 2023, study have been published “Evaluation of point-of-care multiplex polymerase chain reaction in guiding antibiotic treatment of patients acutely admitted with suspected community-acquired pneumonia in Denmark: A multicentre randomised controlled trial” how can you justify the study and compare to this recent study in Denmark.

Comments on the Quality of English Language

Minor changes required.

Reviewer 2 Report

Comments and Suggestions for Authors

I read this study was great interest. It focused the implication of new pathogen identification technic in the management of pneumonia. However, there are several major concerns hinder the acceptance of this study.

1. As PCR test is much faster than routine diagnostic testing, bacteria culture especially. There is no need to carry out a study to prove the shorted time to “target antimicrobial therapy”.

2. The accuracy of PCR test in identification of pathogen is not provided. According to our experience, contamination was very common in PCR test, some other confounding factors were existed. It always leads to unnecessary antibiotic use if rely on PCR test alone. Moreover, if the PCR test is negative, it can hardly conclude that the infection is not exist.

3. Finally and the most importantly, it is unknown that whether patients are benefit from PCR test. Comparison of clinical endpoints between groups, 28-day mortality usually, is needed.

Reviewer 3 Report

Comments and Suggestions for Authors

I thank the authors for the opportunity to read this interesting manuscript.

First of all, I want to congratulate you for the work you have done. I will allow myself to make some small comments.

It would be advisable for the authors to indicate how they have established the sample size calculation to ensure the validity of the study.

I think that one more column should be added to table 1 with the data for the total set of patients.

The authors should indicate the data on mortality, time of admission to the ICU and time of hospital admission. Although it is indicated that the number of patients was limited to draw conclusions about these data, they should be communicated in the paper and allow readers to draw their conclusions.

Round 2

Reviewer 1 Report

Comments and Suggestions for Authors

The authors have made all the required changes.

Comments on the Quality of English Language

Minor changes. A careful read is required.